# Quantifying Antibiotic Distribution in Solid and Liquid Fractions of Manure Using a Two-Step, Multi-Residue Antibiotic Extraction

**DOI:** 10.3390/antibiotics11121735

**Published:** 2022-12-01

**Authors:** Carlton Poindexter, Andrea Yarberry, Clifford Rice, Stephanie Lansing

**Affiliations:** 1Department of Environmental Science and Technology, University of Maryland, College Park, MD 20742, USA; 2Sustainable Agricultural Systems Lab, U.S. Department of Agriculture, Agricultural Research Service, Beltsville Agricultural Research Center, Beltsville, MD 20705, USA

**Keywords:** UPLC-MS/MS, dairy manure, tetracyclines, β-lactams, sulfonamides, macrolides

## Abstract

Antibiotic distribution and analysis within liquid and solid fractions of manure are highly variable due to each compound’s respective physiochemical properties. This study developed and evaluated a uniform method extracting 10 antibiotics from 4 antibiotic classes (tetracycline, sulfonamides, macrolides, and β-lactam) from unprocessed manure, solid–liquid separated manure, and composted solids. Through systematic manipulation of previously published liquid chromatography tandem mass spectrometry methods; this study developed an extraction protocol with optimized recovery efficiencies for varied manure substrates. The method includes a two-step, liquid-solid extraction using 10 mL of 0.1 M EDTA-McIlviane buffer followed by 10 mL of methanol. Antibiotics recoveries from unprocessed manure, separated liquids, separated solids, and heat-treated solids using the two-step extraction method had relative standard deviations < 30% for all but ceftiofur. Total antibiotic recoveries were 67–131% for tetracyclines, 56% for sulfonamide, 49–53% for macrolides, and 1.3–66% for β-lactams. This is the first study to use one protocol to assess four classes of antibiotics in liquid and solid manure fractions. This study allowed for more precise risk assessment of antibiotic transport in manure waste stream applied to fields as a liquid or solid compost.

## 1. Introduction

The practice of antibiotic administration for livestock welfare can leave as much as 90% of the administered antibiotics excreted unmetabolized or as biologically active metabolites in animal manure and urine [1]. The presence and release of antibiotics within agroecosystems influences the development of global antibiotic resistance. Recent legislation within the United States (US) has regulated the use and distribution of medically important veterinary antibiotics [2,3]. While more responsible administration of antibiotics is on the rise, there is a need to determine how antibiotics that are used persist through current manure management treatment systems in place at livestock farms.

Though there are numerous manure management technologies, solid–liquid separation has been increasingly adopted by larger US dairy farms over the past 10 years [4]. This process can produce multiple effluent streams with unique physical/chemical properties that influence antibiotic concentrations and distribution. Rico et al. [5] reported that the liquid fraction of manure contained 54% of the total solids (TS) and 48% of the volatile solids (VS) of the raw manure. The changes in the manure properties due to management and treatment affect the interactions between the manure and antibiotic residuals, solubility, and microbial degradation, which subsequently influences fate and transport of antibiotics, antibiotic resistant genes, and antibiotic resistant bacteria in the environment [4]. Due to the variability in manure composition before and after manure treatment, it is necessary to develop an antibiotic extraction method that is designed to extract multiple antibiotics from manure matrices with vastly different solids concentrations to maintain a baseline metric for comparison. While numerous multi-class antibiotic residual studies have focused on antibiotic extraction and quantitation from manure [6,7,8,9], each of these methods have been fine-tuned for a specific type of manure; either manure solids [8,9,10,11,12,13,14,15,16], manure liquids [7,17,18,19,20], or manure slurries [21,22,23]. The few studies that have investigated multiple manure substrates have used different extraction protocols for the liquid and solid manure fractions [4,21,24,25,26]. The ability to assess antibiotics concentrations in the liquid and solid fractions of manure allows for the capacity to detect both dissolved and sorbed antibiotic residues for more precise risk assessment. Prior to this study, a one-size-fits-all method to extract antibiotics from both liquid and solid manure had not been developed.

Classes of antibiotics with clinical relevance administered to dairy cows include tetracyclines, macrolides, sulfonamides, and β-lactams. While each antibiotic class exhibits different physicochemical properties, several studies have successfully extracted multiple antibiotic classes from solid or liquid manure (not both) using a single extraction protocol, predominantly focusing on tetracyclines, sulfonamides, macrolides, and quinolones [9,11,12,27,28], while few studies reported on β-lactams [20]. These studies did not extract multiple antibiotic classes from both liquid and solid manures using the same protocol. A study by Wang et al. [20] concluded that 94–99% of the antibiotics detected in swine manure were in the solid fraction, with sulfonamides predominantly found in the liquid fraction, yet, they used separate techniques to assess these fractions. To account for the fact that antibiotic concentrations tend to partition based on the solids in the manure, the current study based the sample extraction mass on the total solids (TS) measurements of the manure for the varied manure substrates, which accounted for settled and dissolved solids, as the presence of organics affects antibiotic retention.

To date, there have not been studies that have successfully conducted a single extraction protocol for multiple antibiotic classes that included β-lactams and manure substrates with varying TS concentrations. In this study, several extraction procedures were systematically evaluated to determine the most reliable method for multi-residue antibiotic extraction from four manure types within an on-farm manure management system. The developed protocol created an extraction method for the detection and quantification of four β-lactams, one sulfonamide, three tetracyclines, and two macrolides using ultra performance liquid chromatography tandem mass spectrometry (UPLC-MS/MS) in raw manure, solid and liquid separated manures, and heat-treated manure solids. This work provides further understanding of antibiotic distribution throughout a manure management system, allowing for a more holistic interpretation of the environmental impact of manure utilization on fields as a fertilizer.

## 2. Results and Discussions

### 2.1. Comparative Analysis of Buffer vs Solvent Extraction Efficacy

In this extraction trial, recovery efficiencies of the EDTA-McIlvaine buffer and methanol: acetonitrile (MeOH:ACN) were compared. The recoveries of antibiotics in the cleaned-up EDTA-McIlvaine fraction were between 66% and 88% for the tetracyclines, between 0% and 50% for the β-lactams, 44% for SUL, and 34% and 42% for TUL and TYL, respectively (Figure 1). The recoveries for samples extracted using 50:50 MeOH:ACN were 4 to 15% for the tetracyclines, 0 to 33% for the β-lactams, 69% for SUL, 21% for TUL, and 45% for TYL, respectively (Figure 1). Separate analysis of the extractants indicated strong preferences of some of the antibiotics to either the EDTA-McIlvaine buffer or the organic solvent. Thus, it was decided that a phased two-step extraction procedure would be the most beneficial for the suite of antibiotics in this study. This trial indicated that analytes with higher log K_OW_ coefficients (Table 1) generally exhibited lower recoveries in the aqueous buffer extraction solvent, seen below in Figure 1.

### 2.2. Initial Two-Step Extraction with Buffer Followed by Methanol

In this trial, each manure sample was first extracted using EDTA-McIlvaine followed by extraction using MeOH. In this initial extraction method, the two extracts were analyzed individually to show how much of each antibiotic was recovered by each solvent. When the results for each extraction were combined, recoveries varied between 12 and 77% for all antibiotics (Figure 2), with lower recoveries for the two β-lactams (AMP 12% and CEF 19%) and higher values for PEN (77%). Previous studies have found it difficult to extract β-lactam antibiotics due to the unstable β-lactam ring that is easily hydrolyzed in samples with high bacterial activity [9]. This initial two-step method was successful for the β-lactam PEN and metabolite BEN, which may be due to the separation of the two extraction solvents that operate on different physicochemical properties of the PEN. Kheirolomoom et al. [34] conducted a Penicillin G stability study and found this compound to be the most stable at pH values ranging from 5 to 8 under laboratory conditions. The EDTA-McIlvaine buffer solution used for the extraction had a pH of 5, which may be favorable to the stability of PEN and BEN within the manure matrix. Furthermore, the inclusion of hexane in the extraction protocol to reduce matrix interference [12] proved inefficient (results not shown), thus resulting in the exclusion of this step in the subsequent experiments.

### 2.3. Two-Step Extraction on Solid and Aqueous Manure Samples Based on Total Solids 

The initial two-step extraction from above, without the hexane step, was used as the extraction method for manure collected at four different points at a NY Farm: (1) unprocessed manure pit (UPM), (2) separated liquid manure pit (SL), (3) separated solids (SS) from the top of a screw press, and (4) solids heat treated using a bedding recovery unit (BRU). The average antibiotic recoveries over the four manure types for the EDTA-McIlvaine extraction fraction for the antibiotics that were recovered were 10.5–62.4%, and the average recoveries for the MeOH extraction fraction were between 2.3 and 31.3%. Ampicillin was not recovered from any of the manures using this extraction. The total antibiotic recoveries for the four manure types were calculated by summing the recoveries over the two extraction fractions with average total recoveries for each antibiotic class across the four manure types of: 24 ± 12% to 76 ± 3% for the β-lactams, 50 ± 3% for SUL, 56 ± 2% to 74 ± 2% for the tetracyclines, and 51 ± 5% to 59 ± 9% for the macrolides (Figure 3).

While most of the antibiotics in the SL manure were extracted in the first extraction step using the EDTA-McIlvaine buffer (>67% for all antibiotics), the extraction recoveries for manure with higher solids content ranged from 22–96%, with an average of 63 ± 3.6%. The heightened recovery in the first extraction step (EDTA-McIlvaine buffer) for the SL manure over the other manure types indicates that the antibiotics in the SL manure are mainly in the aqueous phase and not associated with any remaining fine solids in the sample; even though the TS in all samples was held constant (~0.25 g TS/g manure) when samples were prepared for extraction (Table 2). The RSDs for the recoveries of all antibiotics across the four manure types were 3–21% except for CEF (RSD 99%) and TYL (RSD 30%), indicating that this initial two-step method was appropriate as a one-size-fits all extraction approach for most of the antibiotics in this study. The reduced recoveries of CEF and TYL in the UPM and LS samples could be due to the rapid degradation of CEF and TYL in the presence of microorganisms, which would be present in higher bacterial loads in the liquid samples compared to the treated manures [8,15]. Even though CEF and TYL had reduced extraction efficiencies for the more liquid substrates, average recoveries for all antibiotics in all samples were >32%, except CEF (4–46%) and AMP (not recovered). Lower recovery for CEF and AMP is likely due to β-lactam antibiotics being unstable in manure matrices [9]. The performance of this single extraction method on a range of manure types was consistent for most antibiotics and indicates that basing the mass of manure extracted on the TS concentration was a viable sample preparation approach and eliminates the need for multiple antibiotics extraction methods for different manure matrices moving forward.

### 2.4. Optimizing the Two-Step Method to Combine Extracts for One Injection

This set of extractions was conducted to determine the most effective way to combine the two extraction fractions using either Method A or Method B (Figure 4). To compare the extract combination method, the volumes of EDTA-McIlvaine and MeOH were kept at 10 mL in both treatments. The results of the extraction comparison showed that Method A had a significantly higher average recoveries (54%), except for AMP and BEN, which had consistently lower recoveries throughout this study. Recoveries of SUL and the tetracycline drugs were all significantly higher using Method A (extracts combined before SPE) than using Method B (EDTA-McIlvaine fraction cleaned up via SPE and eluted into MeOH fraction) (*p*-values < 0.014), while the macrolide recoveries were not significantly impacted by the extract combination method, and the β–lactam recoveries were drug-dependent (Table 3).

SUL recoveries in Method A (56 ± 3%) were within reported recoveries from manure (64%) and lagoon wastewater (54%) using SPE for cleanup [21]. SUL recovery was likely highest when combining and diluting the extracts prior to SPE due to its hydrophobicity, which acted as a driving force for SUL to sorb to the SPE cartridge resulting in a higher recovery. 

Tetracycline drug recoveries were highest (67–131%) in Method A, which combined and diluted the extracts prior to SPE. The recoveries from Method A were 65%, 41% and 19% higher for OXY (131 ± 13%), TET (114 ± 7%), and CHL (67 ± 3%) in comparison to Method B, respectively (*p*-values < 0.003) (Table 3). Utilization of high volumes of EDTA-McIlvaine buffer has been shown to increase tetracycline recoveries; as a result of the increased concentration of chelators in the solution binding to the cationic solids in the manure and preventing tetracyclines from adsorbing to them [4]. Recoveries of tetracyclines in Method A (67–131%) were within values reported for extraction in manure using liquid-solid extraction followed by SPE (96–170%) [35]. The macrolide recoveries were not significantly impacted by the extract combination method. The lower recoveries for TUL in this study are likely due to the non-extractable portion of antibiotics irreversibly bound to the manure matrix compared to surface water. Jansen et al. [8] observed similar TYL recoveries (~60%) to this study in spiked manure extracted using trifluoric acetic acid in ACN and cleaned up using SPE. Antibiotic interaction within manure differs in comparison to soils or other aqueous matrices due to a larger concentration of natural organic matter, which can provide more sites for sorption [1,4].

The β–lactam recoveries, with exception of PEN, were low throughout the study, which was not surprising, considering β-lactams have proven difficult to recover in previous studies [8,36]. While CEF recovery was significantly higher in Method A than in Method B (*p*-value < 0.001), AMP and BEN recoveries were significantly lower (*p*-value < 0.04) in Method A than in Method B, and the recovery of PEN was not significantly affected by either of the combination methods (*p*-value = 0.121). β-lactams still prove to be a challenging class of analytes. Consistent recovery of PEN using this method is a new development not yet reported in previous extraction studies and is likely due to separation of the solvents during extraction. Since β-lactams (penicillins and cephalosporins) are one of the most used therapeutic drugs for dairy operation (second to tetracyclines) [37], the ability to track penicillin through manure treatment systems and determine their fate in the environment is important. The final two-step extraction using 10 mL each of EDTA-McIlvaine buffer and MeOH and combined with Method A is the recommended method that was proven to be a valuable protocol for multiclass antibiotic analysis and beneficial for the ability to recover penicillin consistently.

### 2.5. Extraction Method Performance Using Method A

The percent recovery, RSD, matrix effect (ME), linearity, limit of detection (LOD), and limit of quantitation (LOQ) were calculated for this method (Table 4). Using the recommended extraction method, acceptable recoveries (>50%) for most antibiotics were shown when spiked at a concentration of 350 µg/kg. The RSD calculated for the recovery study was < 25% for all analytes. The ME ranged from 57 to 89% for all antibiotics, which indicates major signal suppressions from manure extractions for all analytes except TUL due to residual material in the samples. The MEs in these samples were corrected by using an IS for each drug class and labelled analogs, when available, except for β-lactams. The linearity was >0.99 for most analytes obtained from a concentration range of 0.01–1 μg/mL. The LDLs ranged from 0.229–8.05 µg/kg wet weight, and the LOQs ranged from 0.694–24.4 µg/kg wet weight, which illustrates the ability of this method to detect low concentrations of antibiotics in manure.

## 3. Materials and Methods

### 3.1. Standards and Reagents

Reference standards used in the extraction experiments encompassed four groups of antibiotics: tetracyclines, sulfonamides, macrolides, and β-lactams. Standards purchased from Sigma-Aldrich (St. Louis, MO, USA) included tetracycline hydrochloride (TET) at >95%, oxytetracycline hydrochloride (OXY) at >95%, sulfadimethoxine (SUL) at >98.5%, tulathromycin A (TUL) at >95%, tylosin tartrate (TYL) at >80%, penicillin G sodium salt (PEN) at 96–102%, benzylpenicilloic acid disodium salt (BEN) at >95%, ampicillin (AMP) at >95%, and ceftiofur hydrochloride (CEF) at >95%. Chlorotetracycline hydrochloride (CHL) was a United States Pharmacopeia (Rockville, MD, USA) reference standard. At least one internal standard was used for tetracyclines, the sulfonamide, and the macrolides, with labeled standards used when possible (Table 1). No internal standard was used for β-lactams, which were calibrated externally. Demeclocycline hydrochloride (DEM) at 92.4%, a European Pharmacopeia (Strasbourg, France) reference standard, was used as the internal standard for the tetracyclines. Sulfadimethoxine-d6 (SUL-d6) at >99% was used as the internal standard for SUL, roxithromycin (ROX) at >90% was used as the internal standard for TYL, and nalidixic acid-d5-(ethyl-d5) (NAL) at >99% was used as the internal standard for TUL; all purchased from Sigma-Aldrich, St. Louis, MO, USA. High-performance liquid chromatography (HPLC) grade methanol (MeOH) and acetonitrile (ACN), formic acid (88%), oxalic acid (99^+^%), ethylenediaminetetraacetic acid disodium salt dihydrate (EDTA) at 99^+^%, and sodium phosphate dibasic heptahydrate (Certified ACS Crystalline) were purchased from Fisher Scientific (Fairlawn, NJ, USA). Anhydrous citric acid (99%) was purchased from Sigma-Aldrich (St. Louis, MO, USA). Organic-free water was obtained from a Picotech UV Plus system (Hydro Service and Supplies, Gaithersburg, MD, USA). EDTA-McIlvaine (pH 5) solution used for extraction was prepared by combining 500 mL of 0.1 M Citric Acid, 312 mL of 0.2 M disodium phosphate, and 30.25 g EDTA in a 1 L volumetric flask, with organic-free distilled water (DI) used to bring the solution to volume.

Individual stock solutions of reference standards and internal standards (IS) were prepared at a concentration of 100 µg/mL in MeOH and stored in amber glass screw-capped vials at −20 °C and replenished every 6 months. A mixed solution of reference standards for spiking was prepared fresh on the day of the experiment at a concentration of 5 or 7 µg/mL in MeOH. On the day of analysis, the mixed reference standard was prepared at 1 or 5 µg/mL in 50:50 ACN:DI, and a mixed solution containing the internal standards was prepared at a concentration of 10 µg/mL in MeOH.

### 3.2. Collection of Dairy Manure Samples

A blank dairy manure (BDM) sample was collected at the Beltsville Agricultural Research Center (Beltsville, MD USA) farm from dairy cows that were free of antibiotics and included in the extraction experiments. Dairy manure substrates from a farm located in New York, US that used solid/liquid separation, with the liquid manure sent to a lagoon and the separated solids subsequently heat treated using a bedding recovery unit (BRU), were collected and used to compare extraction efficiencies from different manure types [38]. Manure was collected at four points in the bedding recovery treatment process: (1) unprocessed pit manure (UPM), (2) separated liquid manure (SL), (3) separated solids (SS) from the top of a screw press, and (4) solids heat treated using the bedding recovery unit (BRU). After collection, all manure types were stored at 4 °C for no longer than two weeks before extraction.

### 3.3. Sample Preparation

Due to the varying sorptive properties of the different antibiotics extracted in this study (Table 1), the mass of extracted manure was based on the TS concentration (dry matter content) of each manure type, not the total wet mass, that has been used in other studies. The TS concentrations of the manure were determined using the Standard Methods for the Examination of Water and Wastewater [39], with the TS values of each manure substrate tested as shown in Table 2. In brief, this method dries 10 to 25 g of homogenized sample in tared ceramic crucibles at 105 °C overnight, accounting for the dissolved and settled solids masses. Previous studies have extracted antibiotics from 1 to 2.5 g raw manure [13,15,28,40,41,42,43] and from 0.1 to 5 g dried material [4,10,21,23,27]. For this study, the wet mass of the manure fractions corresponded to a dry matter TS content of 0.25, which ranged from 2 to 0.77 g wet weight (2.00 g for BDM; 3.61 g for UPM; 4.20 g for SL; 0.770 g for SS; and 0.722 g for BRU manure substrates).

For the extraction, triplicate manure samples were weighed and, when required, antibiotic standard solutions were added to achieve a concentration of 350 µg/kg of each analyte, following the procedure used by Jansen et al. [8]. All samples were then vortexed for 30 sec and allowed to equilibrate in the dark at room temperature for one-hour prior to extraction. After equilibration, 10 mL of EDTA-McIlvaine buffer was added to each sample. The samples were vortexed for 10 sec, then sonicated in an ultrasonic bath (Elmasonic E 100 H, Singen, Germany) at room temperature for 15 min. Following sonication, the samples were placed on a rotary mixer (Lab-Line Orbit Shaker Model 3520, Melrose Park, IL, USA) and mixed at 50 rpm at room temperature for 15 min. After mixing, the samples were centrifuged at 3300× *g* for 20 min. The supernatant was decanted into a fresh 50 mL polypropylene centrifuge tube. The extraction process was then repeated with 10 mL MeOH. The extracts were either combined or kept separate as described in the various extraction iterations below in Section 3.6.

### 3.4. Sample Cleanup

Throughout the extraction trials, a Phenomenex 33 μm polymeric reverse phase 200 mg/6 mL SPE cartridge (STRATA™-X, Torrance, ON, Canada) was used for sample cleanup. Before use, the cartridge was conditioned with 5 mL MeOH followed by 5 mL EDTA-McIlvaine buffer. The samples were continuously loaded onto the cartridges at a pressure of 10 psi (6 mL/min). After sample loading was complete, the cartridges were washed with 5 mL of DI water and dried by applying vacuum for 5 min. The residues retained on the column were eluted from the cartridge with 5 mL of MeOH into 15 mL polypropylene tubes and subsequently evaporated to <200 μL under a steady stream of N_2_ (10 psi) at 45 °C (Caliper Life Sciences TurboVap LV, Charlotte, NC, USA). Residues were reconstituted to a final volume of 1 or 1.5 mL with 50:50 ACN:DI, vortexed briefly, filtered through a 0.45 μm polytetrafluoroethylene (PTFE) syringe filter, and then analyzed immediately or placed in a −20 °C freezer until analysis on the UPLC-MS/MS (Waters, Milford, MA, USA).

### 3.5. UPLC-MS/MS Conditions

The analysis of antibiotics in manure was conducted using a Waters Acquity H-Class Plus ultra-performance liquid chromatograph (UPLC) tandem Xevo TQ-S micro triple quadrupole MS (Waters Corp. Millford, MA, USA). Chromatographic separation was performed on a Kinetex C-18 (2.6 µm 100 Å, 2.1 mm × 100 mm) column (Phenomenex, Torrance, CA, USA) protected by a C18 guard column. The mobile phase composition was 1 mM oxalic acid and 0.1% formic acid in DI water (mobile phase A) and 0.1% formic acid in ACN (mobile phase B). The analytes were eluted at a flow rate of 0.4 mL/min under the following gradient: 0.0–0.5 min 95:5 (A:B); 0.5–0.7 min linear increase to 35:65 (A:B); 0.7–2.0 min hold; 2.0–2.5 min linear increase to 13:87 (A:B); 2.5–4.0 min linear increase to 100% B; 4.0–6.0 min hold; then 6.0–7.0 min linear decrease to initial 95:5 (A:B); and 7.0–10.5 min hold. Detection was performed in positive electrospray ionization mode (ESI+) with multiple-reaction monitoring (MRM), where each compound was optimized for cone voltage and collision setting. The injection volume was 5 μL and column temperature was held at 40 °C. The MS/MS conditions were as follows: capillary voltage (0.5 kV), source temperature 150 °C, desolvation temperature 450 °C, cone gas flow 50 L/hr, and desolvation gas flow 1000 L/h. Analyte retention times and mass transitions are presented in Table 1. Quantitation was conducted using TargetLynx software (Waters, Milford, MA, USA).

### 3.6. Method Trials 

#### 3.6.1. Comparative Analysis of Buffer vs. Solvent Extraction Efficacy

To determine the efficacy of an aqueous buffer compared to an organic solvent on the extraction of the antibiotics in this study, an experiment was conducted that used either EDTA-McIlvaine buffer or a 50:50 blend of MeOH:ACN. These solvents were tested using six control dairy manure samples that were each spiked with 350 µg of the 10 antibiotics/kg of manure. For one set of triplicate spiked samples 0.1 M EDTA-McIlvaine buffer (pH-4) (10 mL) was used for the extractant, and for the other set of triplicate spiked samples, 50:50 MeOH:ACN (10 mL) was used. The manure was extracted twice as outlined in our methods except that prior to further handling each extract was mixed with 10 mL of hexanes on the rotary mixer (20 rpm, 15 min), following a method used by Pan et al. [12] to help remove interfering fats. After centrifugation, the upper hexane layer was removed and discarded. To limit unwanted impurities, the EDTA-Mcllvaine extract was additionally cleaned up by passing it through SPE and the sorbed antibiotics were eluted off with 50:50 MeOH:ACN. Each extract was concentrated down to <200 µL using a stream of N_2_ (45 °C) and reconstituted to a volume of 1 mL using 50:50 MeOH:ACN prior to analysis using UPLC-MS/MS. Recoveries were calculated based on the concentrations of antibiotics spiked into the manure.

#### 3.6.2. Initial Two-Step Extraction with Buffer Followed by Methanol

This extraction trial employed the extraction solvents stepwise, with aqueous EDTA-McIlvaine used first, followed by MeOH. The use of a two-step protocol over a mixed extractant method was based on the preliminary data from above (Section 3.6.1). For the two-step method, 2 g of the BDM substrate was weighed out in quadruplicate, and three of the samples were spiked with 350 µg/mg antibiotics, with an additional sample carried through as a blank for post-extraction spiking and recovery calculations to account for matrix effects. All samples were placed in the dark for 90 min after spiking. The manure was first extracted using EDTA-McIlvaine (10 mL) following the mixing and centrifugation procedures detailed in Section 3.3. After the EDTA-McIlvaine buffer was decanted into a fresh 50 mL centrifuge tube, MeOH (10 mL) was added to the previously extracted manure pellet for the second extraction. The two extraction fractions were kept separate, and hexanes (10 mL) were added to each fraction, mixed on a rotary mixer (50 rpm, 15 min), and then centrifuged (3300× *g* for 20 min). The EDTA-McIlvaine fraction was further cleaned up using SPE (Section 3.4), with MeOH as the elution solvent. These extracts were prepared for injection on the UPLC-MS/MS, as described above in Section 3.4. The two extraction fractions were analyzed separately, and recoveries for each fraction were calculated based on the response of the post-spiked extract. Total recoveries were calculated by summing the recovery for the two extraction fractions.

#### 3.6.3. Two-Step Extraction on Solid and Aqueous Manure Samples Based on Total Solids

This method was applied on manure collected from a commercial NY dairy farm utilizing a bedding recovery unit (BRU) for the separated solids. The manure samples were collected at each of the four points in the processing line. In lieu of drying samples prior to extraction, extraction masses were based on the TS content of each of the manure types (Table 2). Samples were weighed on a wet basis, according to the TS concentration to keep the solids content consistent across each sample type. The manure samples were weighed out in quadruplicate. Then, three samples of each manure type were spiked to a final antibiotic concentration of 350 µg/kg manure prior to extraction, and one sample was left as a blank for determining recoveries after post-extraction spiking. The four manure-types were extracted using the initial two-step extraction method discussed above in Section 3.6.2 (without the hexane cleanup step). 

#### 3.6.4. Optimizing the Two-Step Method to Combine Extracts for One Injection

A final extraction experiment was conducted to adapt the method employed in Section 3.6.3 above by optimizing it so that the EDTA-McIlvaine and MeOH fractions could be combined prior to analysis on UPLC-MS/MS. This then negates the need to have two extracts to analyze for each sample. In this experiment, two extract combination methods were tested: Method A and Method B (Figure 4). In both Method A and Method B, the manure was extracted as outlined in Section 3.3, the extract fractions were then treated differently prior to being combined. In Method A, the two extraction fractions (10 mL of EDTA-McIlvaine and 10 mL of MeOH) were decanted into the same 50 mL centrifuge tube, the volume was brought up to 30 mL with DI water, centrifuged (20 min at 3300× *g*) to remove suspended matter, and decanted into 470 mL of DI water prior to sample cleanup using SPE (Figure 4A). In Method B, the 10 mL of EDTA-McIlvaine buffer extract was cleaned up using SPE and the 10 mL MeOH extract was blown down to a volume of 5 mL under N_2_, and then it was combined with the SPE eluant prior to the final drying step. These combined extracts were prepared for injection on the UPLC-MS/MS as described above in UPLC-MS/MS section (Section 3.4). The data were reported as averages of triplicate samples and recoveries were calculated based on triplicate blanks that were spiked post-extraction. Statistical comparisons of recoveries for each antibiotic were made between the five treatments using a one-way ANOVA followed by Tukey–Kramer’s post hoc test in R (R Core Team. (2020) Vienna, Austria). The final, optimized method was selected based on the results from careful comparison of Method A and Method B (Figure 4) using replicate spikes of the blank manure.

### 3.7. Evaluation of Performance of Method Trials

Method performance for the method trials were used to establish data reliability based upon linearity of calibrants in the matrix, selectivity, recovery, and repeatability [44]. Matrix linearity was determined for each analyte by adding solutions of the antibiotics into the blank manure matrix at five concentrations from 10 ng/mL to 1 µg/mL. Least squares regression was conducted on the calibration lines constructed by plotting the ratio of the peak area of the analyte to the area of the internal standard (IS) against the ratio of the added concentration of analyte to the IS. A correlation coefficient greater than 0.99 was considered linear. Method limit of detection (LOD) (Equation (1)) and limit of quantitation (LOQ) (Equation (2)) were determined based on the standard deviation (SD) of the response of the lowest calibrant in the matrix-matched calibration curve and the slope of the curve [44]. Selectivity was conducted by comparing the chromatograms of the blank manure and spiked manure to ensure there was no interference in the blank sample at any of the antibiotic ion transitions. Recovery quantitation was based on the concentrations of analytes in the blank matrices fortified before and after extraction (Equation (3)). Repeatability was assessed by using the relative standard deviation (%RSD) of triplicate samples during one run (Equation (4)). Matrix effect (ME) was evaluated by comparing the slope of the calibration curve in the matrix extract to the slope in the curve prepared in solvent (Equation (5)) [11].
(1)$$LOD=\frac{3.3\times SD}{slope}$$
(2)$$LOQ=\frac{10\times SD}{slope}$$
(3)$$Recovery\;\left(\%\right)=\frac{Concentration\;in\;pre-spiked\;sample}{Concentration\;in\;post-spiked\;sample}\;x$$
(4)$$\%RSD=\frac{SD\;of\;the\;mean}{mean}\times 100$$
(5)$$ME\;\left(\%\right)=1-\frac{Slope\;of\;curve\;in\;extract}{Slope\;of\;curve\;in\;solvent}\times 100$$

## 4. Conclusions

This study and the final method presented builds on traditional extraction methodology and contributes three major concepts: (1) better understanding of solvent/buffer interaction based on manure matrix and antibiotic type, (2) an extraction process based on the sample TS, and (3) a uniform method for extraction of four antibiotics classes that can be used for the various forms of manure throughout the manure treatment and land fertilization processes. The most reliable method for extracting and analyzing the antibiotics evaluated in this study was a two-step extraction, with the final combined extracts diluted to an organic solvent concentration of 2% and then cleaned up using SPE (Method A). When multiple manure types were extracted, keeping the TS concentration of each manure constant resulted in consistent antibiotic extraction among the manure types. This method of determining the mass of manure to extract for similar recoveries can be used to evaluate antibiotic concentrations in manure as it moves through a treatment/land fertilization system. Furthermore, the assessment of antibiotic concentrations in multiple manure treatment effluent streams, with liquid and solid fractions, is vital to understanding antibiotic contamination risks between substrate type and final fertilizer applications using a mass balance approach.

## Figures and Tables

**Figure 1 antibiotics-11-01735-f001:**
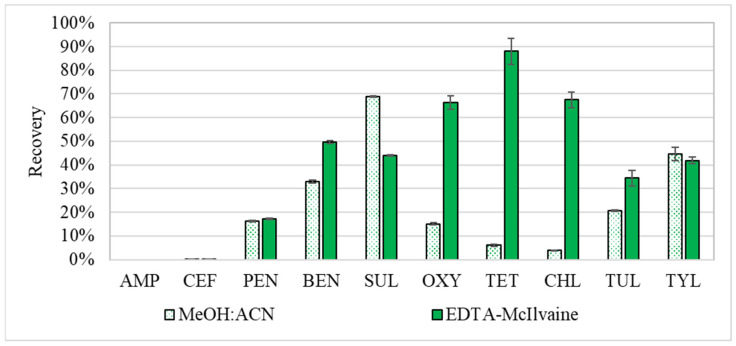
Recoveries from extraction of spiked manure with either EDTA McIlvaine buffer (solid bar) or 50:50 MeOH:ACN (the dotted bar). Recoveries are treatment averages (*n* = 3), with error bars based on ± standard error. The antibiotics examined include Ampicillin (AMP), Ceftiofur (CEF), Penicillin-G (PEN), Benzylpenicilloic Acid (BEN), Sulfadimethoxine (SUL), Oxytetracycline (OXY), Tetracycline (TET), Chlortetracycline (CHL), Tulathromycin-A (TUL), and Tylosin Tartrate (TYL).

**Figure 2 antibiotics-11-01735-f002:**
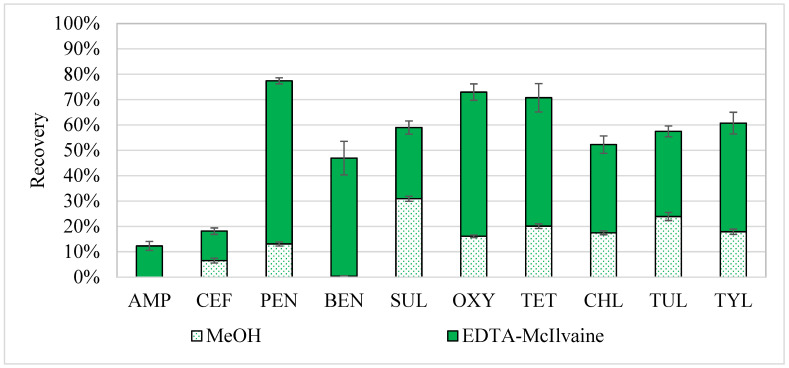
Recoveries from the initial two-step extraction experiment with EDTA-McIlvaine (solid bar) as the first extractant followed by MeOH (dotted bar) as the second extractant. Resulting recoveries were averages (n = 3), with error bars based on ± standard error. The antibiotics examined included Ampicillin (AMP), Ceftiofur (CEF), Penicillin-G (PEN), Benzylpenicilloic Acid (BEN), Sulfadimethoxine (SUL), Oxytetracycline (OXY), Tetracycline (TET), Chlortetracycline (CHL), Tulathromycin-A (TUL), and Tylosin Tartrate (TYL).

**Figure 3 antibiotics-11-01735-f003:**
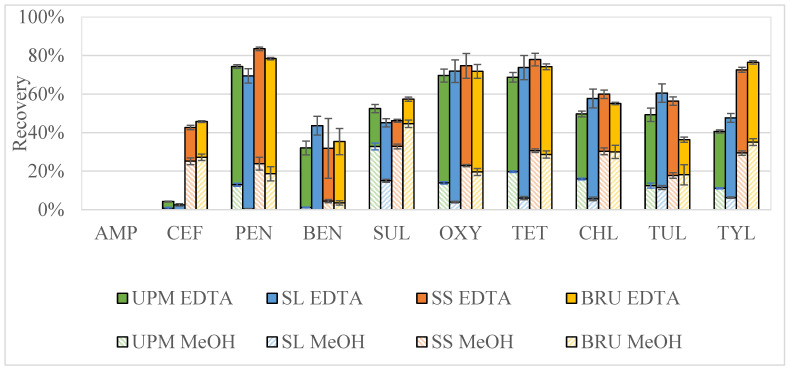
Extraction recoveries of antibiotics from four types of manure processed through a manure treatment system (graphed from left to right within each antibiotic): unprocessed manure (UPM), liquid separated manure (SL), solid separated manure (SS), and manure treated using a bedding recover unit (BRU). The extraction was a two-step process with an EDTA-McIlvaine buffer as the first extractant (EDTA, top half of the bar) and MeOH as the second (bottom half of the bar). Values from each extraction fraction are averages (n = 3) with ± standard error bars. The antibiotics examined include Ampicillin (AMP, not recovered), Ceftiofur (CEF), Penicillin-G (PEN), Benzylpenicilloic Acid (BEN), Sulfadimethoxine (SUL), Oxytetracycline (OXY), Tetracycline (TET), Chlortetracycline (CHL), Tulathromycin-A (TUL), and Tylosin Tartrate (TYL).

**Figure 4 antibiotics-11-01735-f004:**
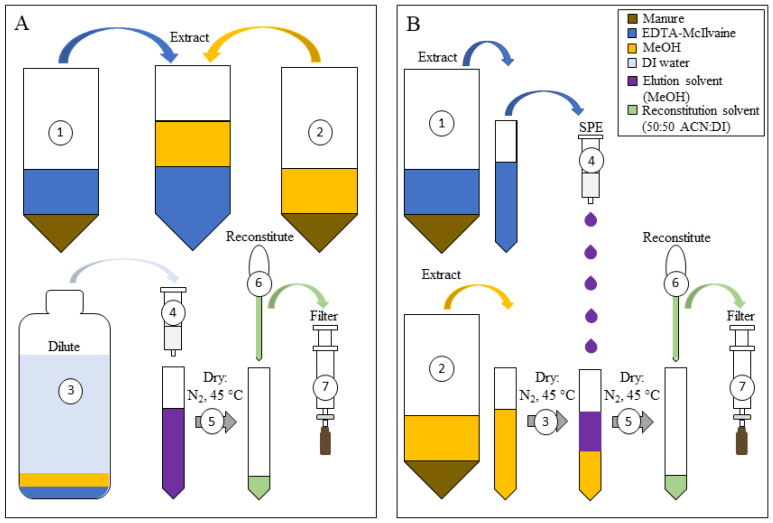
EDTA-McIlvaine and MeOH extracts based on methods in Section 3.6.3 were combined using two methods: (**A**) the two extract fractions were mixed, then diluted to 500 mL with DI water prior to cleanup using solid phase extraction (SPE), and (**B**) the EDTA-McIlvaine fraction was cleaned up using SPE and then eluted into the MeOH fraction. Circled numbers refer to sequence of steps in each method.

**Table 1 antibiotics-11-01735-t001:** Select physiochemical properties and analytical method parameters of antibiotics inves-tigated in this study, including their chromatographic retention times (RT).

Antibiotics	Acronym	Internal Standard	Molecular Weight (g/mol)	Precursor Ion (*m*/*z*)	Product Ion (*m*/*z*)	RT (min)	Log Kow ^b^	pKa
**β-Lactams**								
Ampicillin	AMP	NA	349.41	350.1	106.1 ^a^, 114.0	1.88	1.35	3.07, 7.12 ^c^
Ceftiofur	CEF	NA	523.56	524.0	241.0 ^a^, 125.2	2.03	1.22	2.64, 3.44, 10.7 ^d^
Penicillin-G	PEN	NA	334.39	335.1	217.1 ^a^, 160.0	2.13	1.83	2.97, 4.75 ^c^
Benzylpenicilloic Acid	BEN	NA	352.4	353.1	160.0 ^a^, 128.0	2.12	ND	ND
**Sulfonamides**								
Sulfadimethoxine	SUL	SUL-d6	310.33	311.1	156.1 ^a^, 92.1	2.11	1.63	1.62, 6.13 ^e^
**Tetracyclines**								
Oxytetracycline	OXY	DEM	460.44	461.2	201.1 ^a^, 98.1	1.89	−0.9	3.71, 8.08, 10.15 ^b^
Tetracycline	TET	DEM	444.3	445.0	410.2 ^a^, 154.1	1.89	−1.37	3.56, 7.09, 9.28 ^c^
Chlorotetracycline	CHL	DEM	478.88	479.1	154.1 ^a^, 98	1.94	−0.62	3.49, 7.14, 9.28 ^c^
**Macrolides**								
Tulathromycin-A	TUL	NAL	806.1	403.9	72.1 ^a^, 116.1	1.86	3.69	8.6–9.6 ^e^
Tylosin Tartrate	TYL	ROX	916.112	916.5	174.1 ^a^, 101.0	1.97	1.95	7.71^c^
**Internal Standards**								
Sulfadimethoxine-d6	SUL-d6	NA	316.37	317.95	108.0 ^a^	2.11		
Demeclocycline	DEM	NA	464.86	465.1	154.1 ^a^	1.92		
Nalidixic acid-d5	NAL	NA	237.27	238.24	104.2 ^a^	2.23		
Roxithromycin	ROX	NA	837.06	837.54	158.1 ^a^	2.07		

^a^ Indicates quantitative ion, ^b^ EPA [29], ^c^ Zrncic et al. [30], ^d^ Ribeiro and Schmidt [31], ^e^ Geiser et al. [32], Villarino et al. [33].

**Table 2 antibiotics-11-01735-t002:** Total solids (TS) of the manure used in this study, and the wet mass used in the extraction experiments based on a TS of 0.25 g/g manure for extraction. Solids are reported as average values (n = 3) with ± standard error.

Manure	Total Solids (g_solids_/g_wet manure_)	Mass Extracted (g_wet_)
Blank Dairy Manure (BDM)	0.134 ± 0.005	2
Unprocessed Pit Manure (UPM)	0.074 ± 0.004	3.61
Separated Liquid (SL)	0.064 ± 0.0002	4.20
Separated Solids (SS)	0.347 ± 0.010	0.770
Bedding Recovery Unit (BRU)	0.370 ± 0.120	0.722

**Table 3 antibiotics-11-01735-t003:** Results for optimization experiment that combined extracts for one injection per sample. Method A was compared to Method B as diagrammed in Figure 4.

Antibiotic	Method A	Method B	*p*-Value
	Recoveries (%) ± SD		
Oxytetracycline	131 ± 13	45 ± 4	0.0004
Tetracycline	114 ± 7	68 ± 2	0.0004
Chlorotetracycline	67 ± 3	54 ± 1	0.0021
Penicillin-G	66 ± 1	58 ± 1	0.0006
Sulfadimethoxine	56 ± 3	31 ± 2	0.0003
Tylosin	53 ± 2	47 ± 6	0.18
Tulathromycin-A	49 ± 9	43 ± 2	0.32
Ceftiofur	11 ± 0.3	6.4 ± 0.5	0.0002
Ampicillin	2.3 ± 0.1	7.3 ± 0.4	0.00003
Benzylpenicilloic Acid	1.3 ± 0.04	5.6 ± 0.4	0.0001

**Table 4 antibiotics-11-01735-t004:** Method performance parameters for the two-step extraction of antibiotics from manure using Method A.

Antibiotics	Recovery (%RSD), n = 3	Matrix Effect	Linearity Fit (R^2^)	LOD ^a^ (µg/kg)	LOQ ^b^ (µg/kg)
AMP	2% (4%)	88%	0.995	3.58	10.8
CEF	11% (21%)	88%	0.996	0.893	2.71
PEN	57% (22%)	79%	0.986	2.53	7.68
BEN	0.5% (31%)	85%	0.988	4.83	14.6
SUL	56% (8%)	68%	0.999	0.606	1.84
OXY	131% (17%)	88%	0.999	8.05	24.4
TET	114% (10%)	88%	0.999	2.02	6.11
CHL	66.2% (6%)	89%	0.999	7	21.2
TUL	47% (25%)	57%	0.989	3.18	9.64
TYL	55% (3%)	86%	0.996	0.229	0.694

^a^ Limit of detection, ^b^ Limit of quantitation.

## Data Availability

Data is available within the article.

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
