# Peer review of "Quantifying Antibiotic Distribution in Solid and Liquid Fractions of Manure Using a Two-Step, Multi-Residue Antibiotic Extraction"

_antibiotics, 2022, doi:10.3390/antibiotics11121735_

Round 1

Reviewer 1 Report

It is a study on an important subject.

Author Response

Thanks for the approval of all points,

Reviewer 2 Report

Dear authors,

Thank you for your exciting manuscript. Overall, the manuscript is well organized and presents data and writing. However, there are some points that should be revised before publication.

1. Fig. 1 should appear in numerical order. Why Figs. 2-4 first appeared and then Fig. 1. Fig. 2 should be Fig.1 and so on. Please cite Fig. in the text and ensure that the first citation of each figure appears in numerical order.

2. There are a number of typos in the manuscript. Please carefully checked and revised them in the revision process.

3. Fig. 4 is hard to understand. Please make them into a simplified version or separate graph for easy understanding. 

4. Also, Fig. 1 (schematic) is not clear. Please re-draw it in a higher manner and clear step-by-step of the experiment. 

5. Reference styles are not identified. Please checked and corrected the format of the references based on the journal guideline. 

Author Response

.1  The figures were re-numbered and citations fixed to follow the new order.  Changes are noted in the revised version.

2.  The typos and minor corrections were taken care of and can be observed in the marked up revised version being submitted.

3.  Figure 4 was modified by making the EDTA-Fraction a solid color and the MeOH-Fraction a dotted-version of the same color for each manure type. In the caption, the order of the manure types in the figure was specified to help readers without color versions.

4.  Modified Figure 1 (now) by numbering each step for each method in each of the subplots and also noted this in the figure title.  

5.  Reference style conforms to numbered citation formats found in current publication in Antibiotics manuscripts.  While there were a few incorrect forms which we corrected where instead of author et al. the full list of authors were included and one citations, Pan et al. was converted to the numbered citation format.

Reviewer 3 Report

The manuscript reports the results obtained by developing a single protocol suitable to extract and quantify different classes of antibiotic from both liquid and solid manure fractions. The importance of these results lies in the new possibility to determine by a single procedure the amount of antibiotics' residues in any kind of manure , treated or not, before to use it in agriculture as fertilizer.

The employed methods and protocols are very clearly described and the paper is well prepared. I suggest you to implement just few minor changes to the manuscript.

Title: rather than "interactions" it would be more informative to put different term such as "presence" or "distribution" or similar.

Line 70: check the reference citation.

Figure 4: the style used for this graph make it unclear. I suggest you to employ  for each compound the same light colors but filling one part of each column  with full color and the other one with a grid, the same for all columns.

Figure 1: it can be moved in the R&D section, at the paragraph 2.4, the number of the figures have to be rearranged.

Author Response

Thank you for the compliments.

  1. We agreed with the replacement of interactions and used "distribution" in the new title.
  2. We fixed the citation on line 70 and a few others found now on lines 234 and 347 (marked text) now.
  3. We modified Figure 4, by using solid colors for the EDTA extractions and speckled filled the bars region for the methanol extractions.
  4. We disagree and left Figure 1 (now 4) in the Method and Materials section.  It is a methods treatment description and belongs here.  And, yes we rearranged the numbers for the figures. 

Reviewer 4 Report

It is my pleasure to revie the manuscript entitled “Quantifying antibiotic interactions in solid and liquid fractions of manure using a two-step, multi-residue antibiotic extraction” submitted to the journal Antibiotics. The authors have developed a protocol for extraction of antibiotics in a single extraction method.  The manuscript is well written and data is presented elegantly. The method can be applied for monitoring the antibiotic residue in manure treatment or land fertilization. Following minor change can be made to the manuscript.  

1.    Abstract : Abstract has some of the details that can be moved to Methods section(lines 20-23) .  

Author Response

We appreciate the encouraging words.

1.  We 

Removed the sentences (lines 21-24), “After each extractant addition, the solutions were vortexed (10 sec), sonicated (15 min), rotary mixed (15 min), and centrifuged at 3,300 x g (20 min).”  Decided to leave in sentence on lines 20-21, This sentence defines the method precisely as to solvent choices.